A validation of Illumina EPIC array system with bisulfite-based amplicon sequencing

Noble Alexandra J. alexandra.noble@pg.canterbury.ac.nz 1
Pearson John F. 2
Boden Joseph M. 3
Horwood L. John 3
Gemmell Neil J. 4
Kennedy Martin A. 2
Osborne Amy J. 1
1 School of Biological Sciences, University of Canterbury , Christchurch , New Zealand
2 Department of Pathology and Biomedical Sciences, University of Otago , Christchurch , New Zealand
3 Department of Psychological Medicine, University of Otago , Christchurch , New Zealand
4 Department of Anatomy, Univeristy of Otago , Dunedin , New Zealand
Patnaik Santosh
Electronic publication date: 2021 Feb 10
Publication date: 2021
Volume: 9
Electronic Location ID: e10762
Received 2020 Aug 4; Accepted 2020 Dec 22
Copyright: ©2021 Noble et al.
Copyright year: 2021
Copyright holder: Noble et al.
License: This is an open access article distributed under the terms of the Creative Commons Attribution License, which permits unrestricted use, distribution, reproduction and adaptation in any medium and for any purpose provided that it is properly attributed. For attribution, the original author(s), title, publication source (PeerJ) and either DOI or URL of the article must be cited.
License URL: https://creativecommons.org/licenses/by/4.0/

Keywords: DNA methylation, Illumina EPIC array, Bisulfite based amplicon sequencing

Funding: University of Otago Research The Carney Centre for Pharmacogenomics The Health Research Council of New Zealand Programme Grant 16/600 The Canterbury Medical Research Foundation This study was funded by University of Otago Research Grant to Martin Kennedy, The Carney Centre for Pharmacogenomics. Christchurch Health and Development Study (CHDS) was funded by the Health Research Council of New Zealand (Programme Grant 16/600) and the Canterbury Medical Research Foundation. The funders had no role in study design, data collection and analysis, decision to publish, or preparation of the manuscript.

==============================
The Illumina Infinium® MethylationEPIC BeadChip system (hereafter EPIC array) is considered to be the current gold standard detection method for assessing DNA methylation at the genome-wide level. EPIC arrays are often used for hypothesis generation or pilot studies, the natural conclusion to which is to validate methylation candidates and expand these in a larger cohort, in a targeted manner. As such, an accurate smaller-scale, targeted technique, that generates data at the individual CpG level that is equivalent to the EPIC array, is needed. Here, we tested an alternative DNA methylation detection technique, known as bisulfite-based amplicon sequencing (BSAS), to determine its ability to validate CpG sites detected in EPIC array studies. BSAS was able to detect differential DNA methylation at CpG sites to a degree which correlates highly with the EPIC array system at some loci. However, BSAS correlated less well with EPIC array data in some instances, and most notably, when the magnitude of change via EPIC array was greater than 5%. Therefore, our data suggests that BSAS can be used to validate EPIC array data, but each locus must be compared on an individual basis, before being taken forward into large scale screening. Further, BSAS does offer advantages compared to the probe-based EPIC array; BSAS amplifies a region of the genome (∼500 bp) around a CpG of interest, allowing analyses of other CpGs in the region that may not be present on the EPIC array, aiding discovery of novel CpG sites and differentially methylated regions of interest. We conclude that BSAS offers a valid investigative tool for specific regions of the genome that are currently not contained on the array system.

Introduction

Epigenetic modifications, such as DNA methylation, play a vital role in regulating gene expression (Hackett & Surani, 2013) and have the potential to induce phenotypic changes (Dolinoy, Huang & Jirtle, 2007; Sinclair et al., 2007; Kucharski et al., 2008; Gertz et al., 2012; Wang et al., 2012). DNA methylation occurs when a methyl group is covalently transferred to the C5 position of the cytosine ring of a DNA molecule by a methyltransferase enzyme, with the resulting modified cytosine then termed 5-methylcytosine (5mC) (Mitchell, Schneper & Notterman, 2016). In mammals, most DNA methylation occurs at CpG dinucleotides. CpG sites themselves can be defined as a singular modified cytosine residue which are found throughout the genome, but are particularly dense in promoter regions (Takai & Jones, 2002).

DNA methylation is heavily influenced by the surrounding environment; factors such as tobacco smoking (Breton et al., 2009; Zeilinger et al., 2013; Ambatipudi et al., 2016a; Ambatipudi et al., 2016b; Osborne et al., 2020), alcohol (Philibert et al., 2012; Liu et al., 2016), nutrition (Rampersaud et al., 2000; Delgado-Cruzata et al., 2015), stress (Murgatroyd et al., 2009) and aging (Horvath et al., 2012; Marioni et al., 2015) can all impact on DNA methylation at CpG sites. Alterations to DNA methylation are associated with changes in phenotype and also, in some instances, methylation changes contribute to disease pathology (Kim et al., 2010; Mastroeni et al., 2010; Rakyan et al., 2011; De Jager et al., 2014).

As a result of these relatively recent observations, the assessment of differential DNA methylation in humans, and in particular, epigenome-wide association studies (EWAS), is a burgeoning field. High-throughput array technologies are a popular choice for EWAS, due to their robustness and accuracy (Pidsley et al., 2016). The Illumina Infinium® MethylationEPIC array (hereafter ‘EPIC array’) quantifies methylation at 850,000 different CpG sites (Zhou, Laird & Shen, 2017), and although this is still a small proportion of the total number of CpG sites in the genome (∼28 million Lövkvist et al., 2016) it represents a broad distribution of sites that give a specific and robust measurement of methylation at those sites.

Further, the goal of many whole-genome studies of DNA methylation is often a pilot or scoping study to capture a range of targets that may be associating with, e.g., a particular environmental exposure. As such, once the genome has been investigated in a number of samples, a whole-genome approach is not always necessary if the user simply requires follow up and/or validation of identified loci in a larger cohort. To undertake further analyses and to validate methylation array-based experiments, several different methods exist that that rely on bisulfite treatment of DNA: bisulfite-based amplicon sequencing (BSAS), bisulfite pyrosequencing and methylation-specific PCR (MS-PCR) are methods which can specifically target a predetermined area of interest in the genome at a low cost and higher sample throughput, compared to arrays. An informative study conducted by the BLUEPRINT consortium evaluated 27 predefined genomic regions, across 32 reference samples amongst 18 laboratories using six assays (Bock et al., 2016). Good agreement was observed across methods, with amplicon bisulfite sequencing, and bisulfite pyrosequencing showing the best concordance (Bock et al., 2016). A similar study also assessed bisulfite pyrosequencing, observing congruence to EPIC array analysis (Roessler et al., 2012). However, pyrosequencing is known to have quantitative flaws due to the output of sequences generated through fluorescence methods (França, Carrilho & Kist, 2002). MS-PCR is a method often used in clinical settings (Herman et al., 1996), however it has a high false positive rate (Claus et al., 2012). By contrast, BSAS detects cytosine methylation to base-pair scale resolution without reliance on light detection methods for sequencing (Masser, Stanford & Freeman, 2015). BSAS is a multiplex procedure that can quantitatively assess each CpG site within numerous target regions at the same time (Masser, Berg & Freeman, 2013).

Thus, given the limitations of pyrosequencing and MS-PCR, here we examine whether BSAS can also accurately validate EPIC array data, and be used as a replication, and/or expansion tool for targeted DNA methylation analyses, similar to what has been shown using pyrosequencing. Further, we wish to assess the multiple other CpG sites residing within the targeted amplicon region, to investigate differential methylated regions, which would not be able to be explored via EPIC array.

To answer the question, we used EPIC array data generated from individuals from the Christchurch Health and Development Study (CHDS) which evaluated differential DNA methylation in response to regular cannabis use (Osborne et al., 2020). The CHDS is a longitudinal study of a birth cohort of 1265 children born in 1977 in Christchurch, New Zealand, who have been studied on 24 occasions from birth to the age of 40 (n = 904 at age 40). Of this, a total 96 individuals were selected, and arrays were performed in two separate batches in consecutive years (n = 48 per year).

For validation analysis we selected individuals with EPIC array data (n = 14), as well as new individuals (n = 82), to serve as a validation and expansion cohort for the differential DNA methylation identified via EPIC array (Osborne et al., 2020). Specifically, we asked whether BSAS, after determination of the most appropriate normalisation method, produced the same average methylation values as EPIC arrays, when comparing case data to control data.

While both EPIC array and BSAS are readily used as standalone experiments, they would provide robust evidence if carried out together. Thus, given the rising popularity of studies investigating DNA methylation, establishing a better understanding of how differential DNA methylation differs between regions within the genome, such as evaluating concordance between methods and then further assessing resultant CpG sites within a designated region, is valuable to the scientific community.

Materials & Methods

Cohort selection and DNA extraction - EPIC arrays

EPIC array data used in this study has previously been published (Osborne et al., 2020). Briefly, in this study we use DNA from human participants who are partitioned into three groups: (i) regular cannabis users, who had never used tobacco (“cannabis-only”); those who consumed both cannabis and tobacco (“cannabis plus tobacco”), and; (ii) controls, who consumed neither cannabis nor tobacco. Controls were matched as closely as possible for sex, ethnicity and parental socioeconomic status (data and methods described in Osborne et al., 2020).

Bioinformatics analysis—processing and normalisation of raw EPIC array data

For this study, analysis was carried out using R statistical software (Version 3.5.2) (Team, 2013). Quality control was first performed on the raw data; sex chromosomes and 150 failed probes (detection P value greater than 0.01 in at least 50% of samples) were excluded from analysis. Furthermore, potentially problematic CpGs with adjacent single nucleotide polymorphisms (SNPs), or that did not map to a unique location in the genome (Pidsley et al., 2016) were also excluded. The raw data were then normalised using Noob (Fortin, Triche Jr & Hansen, 2017) in the minfi package (Aryee et al., 2014). Normalisation was then checked by observing density plots as well as multidimensional scaling plots of the 5,000 most variable CpG sites.

Cohort selection and DNA extraction—BSAS experiments

BSAS analysis was carried out on two groups: cannabis plus tobacco users (n = 44) and controls (n = 38), who had never used cannabis. In contrast to the EPIC array analysis, no cannabis-only participants were used in BSAS; this is a consequence of the small number of individuals who use cannabis but who do not also use tobacco. Cannabis users were all selected on the basis that they either met DSM-IV diagnostic criteria (American Psychiatric Association, 1994) for cannabis dependence or had reported using cannabis consumption on a daily basis for a minimum of three years prior to age 28. Participants were matched as closely as possible for the following variables, sex, ethnicity, and parental socioeconomic status (Table S1). All participants in this birth cohort were enrolled across a four month period so they are all of a similar age. Collection and analysis of DNA in the Christchurch Health and Development Study was approved by Southern Health and Disability Ethics Committee (CTB/04/11/234/AM10). DNA extraction protocols were previously described in (Noble et al., 2020). Specifically, DNA was extracted from whole blood samples using a Kingfisher Flex System (Thermo Scientific, Waltham, MA USA) and quantified via nanodrop (Thermo Scientific, Waltham, MA USA). DNA was bisulfite treated using the EZ DNA Methylation-Gold kit (Zymo Research, USA) as per the manufacturer’s instructions.

CpG site selection, primer design and amplification—BSAS

A total of 15 CpG sites, representing 15 individual probes from the Illumina EPIC array, were chosen based on their differential methylation status in cannabis plus tobacco users compared to controls (Table 1). A range of probes at differing levels of significance (not significant, nominally significant, significant after P-value adjustment) were chosen to reflect the range of data provided by the EPIC arrays. Primers to amplify bisulfite-treated DNA were designed using the online tool BiSearch (Arányi et al., 2006) to amplify a ∼250 base pair region which spanned the CpG site (Table S2). At the 5′ end of each primer sequence, an Illumina overhang (33 base pair sequence) was included to ensure the ability to pool the amplicons and barcode them for high-throughput sequencing. All product lengths were all between 226 and 340 base pairs. To ensure primer specificity, Delta G’s were designed to be no lower than -9 kcal/mol for efficiently, using the tool OligoAnalyzer (IDT®). A total of 30 primer pairs were initially designed for this experiment, and 15 of these are discussed here, as these were the primer pairs which performed efficiently at first usage. PCRs were undertaken as per (Noble et al., 2020).

Table 1 CpG site differences from EPIC array and the BSAS methods at the 15 loci of differing levels of significance (not significant, nominally significant, and significant after P-value adjustment).

				Illumina EPIC array	BSAS	Difference between methods	
	Cg/Gene	Position in genome	Illumina ID	β difference	P value	FDR Adjusted P value	β difference	P value	FDR Adjusted P value	β difference	
1	AHRR	Chr5, GB	cg05575921	−0.233	5.33E−12	3.7E−06	−0.041	0.006	0.245	−0.192	
2	cg11977356a	Chr19	cg11977356	−0.040	0.474	0.999	−0.004	0.406	0.959	−0.036	
3	ITPR1	Chr3, GB	cg08987995	−0.001	0.572	0.999	0.005	0.820	0.822	−0.006	
4	MAGI	Chr7, GB	cg21121803	−0.008	0.572	0.999	−0.007	0.809	0.959	−0.0004	
5	EHMT2	Chr6, GB	cg07829740	0.005	0.037	0.999	−0.015	0.071	0.579	0.020	
6	PPM1L	Chr3, GB	cg26406186	−0.006	0.818	0.999	0.011	0.904	0.963	−0.017	
7	cg00571101a	Chr12	cg00571101	0.004	0.368	0.999	−0.004	0.813	0.952	0.008	
8	cg09078959a	Chr5	cg09078959	−0.001	0.893	0.999	−0.005	0.001	0.245	0.004	
9	cg01614625a	Chr7	cg01614625	−0.009	0.370	0.999	−0.006	0.569	0.952	−0.004	
10	DP10	Chr2, GB	cg05868547	0.006	0.077	0.999	−0.003	0.713	0.952	0.009	
11	cg11293828a	Chr12	cg11293828	−0.014	0.665	0.999	0.032	0.735	0.952	−0.045	
12	CHD7	Chr5, 5′UTR	cg19926587	−0.007	0.960	0.999	−0.006	0.429	0.959	−0.001	
13	NIPAL4	Chr5, TSS1500	cg17695979	−0.007	0.714	0.999	−0.003	0.106	0.713	−0.004	
14	PRDM5	Chr4, GB	cg01118724	−0.004	0.734	0.999	0.005	0.116	0.713	−0.009	
15	SLC17A7	Chr19, GB	cg02624701	−0.043	0.312	0.999	0.018	0.646	0.952	−0.061	
Notes.

a When a cg number is listed, then there is no known gene associated with that CpG site. GB-Gene Body.

Following the PCR, DNA was cleaned up with Agencourt® AMPure® XP beads (Beckman Coulter) and washed with 80% ethanol and allowed to air-dry. DNA was then eluted with 52.5 µl of 10 mM Tris pH 8.5 before being placed back into the magnetic stand. Once the supernatant had cleared, 50 µl of supernatant was transferred into a fresh 96-well plate. DNA samples were quantified using the Quant-iT™ PicoGreen™ dsDNA Assay kit (Thermo Fisher) using the FLUROstar® Omega (BMG Labtech). Sequence libraries were prepared using the Illumina MiSeq™ 500 cycle Kit V2, and sequenced on an Illumina MiSeq™ system at Massey Genome Services (Palmerston North, New Zealand).

Bioinformatic and statistical analysis—BSAS data

Illumina MiSeq™ sequences were trimmed using SolexaQA++ software and aligned to FASTA bisulfite converted reference sequences using the package Bowtie2 (version 2.3.4.3). Each individual read was then aligned to all reference sequences (GRCh37/hg19) using the methylation-specific package Bismark (Krueger & Andrews, 2011). Bismark produced aligned mapped reads with counts for methylated and unmethylated cytosines at each CpG site, thus BSAS returns additional CpG sites to the intended validation target, as each sequencing read contains multiple CpG sites. Cytosine proportion is calculated based upon the number of cytosines divided by the number of cytosines with the additions of the number of thymines present (C∕(C1) + T). This gave the average methylation β values for each individual at each given CpG site. These β values ranged between 0 - 1, with a β equal to 1 indicating 100% methylation at that CpG site across all sequencing reads. These data were imported into R Studio (RStudio version 3.3.0) and the edgeR package (Chen et al., 2017) was used to determine differential DNA methylation between cannabis users and controls; coverage level was set to greater or equal to “8” across unmethylated and methylated counts. This was also set at 50 and 100 reads and no differences were seen between the results at any of these thresholds, so “8” was used for the continuation of BSAS calling under the recommendations of Chen et al. (2017) whereby the conservative rule of thumb is total count (both methylated and unmethylated) is at least “8” in every sample. Within the data set 96.5% of the reads were above a methylation coverage of 50 (Data S1). A negative binomial generalised model was used to fit the counts (methylated and unmethylated reads) in regards to the two variable groups. Summary tables compiled of the CpG sites of interest with nominal P value significance and post multiple testing using false discovery rate (FDR) of less than 0.05 were considered to be statistically significant. A scatter plot including a linear regression line with adjusted R2 values was generated in R Studio to quantify the correlation between β values produced with EPIC array and BSAS. Adjusted R2 values were calculated for: (i) BSAS cases versus EPIC cases, and; (ii) BSAS controls versus EPIC controls. A Bland Altman analysis (Bland & Altman, 1986) was used to compare the agreement of the two techniques. Means were log transformed and lower and upper levels of agreement with 95% confidence intervals were calculated. Welch two sample t-tests were carried out on each of the loci (cases and controls separately) to assess differences between the two methods. All graphs were constructed using the R package ggplot2 (Wickham, 2016).

Results

Validation and replication of EPIC array data using BSAS

The differences between average methylation (β values) of cannabis plus tobacco users (cases) and controls were calculated for each method (EPIC array and BSAS, Table 1).

When comparing case vs control data from EPIC and BSAS individually, no significant difference in average methylation between case and control was observed for either detection method, with the exception of cg05575921 in AHRR and the site cg09078959. AHRR was significant in both BSAS and EPIC (P = 0.006, P = 5.33 ×10−12), and cg05575921 was found to only be significant under BSAS (P = 0.001, P = 0.665).

Correlations between BSAS and EPIC were plotted individually for cases and controls. BSAS versus EPIC cases resulted in an adjusted R2 of 0.8878 and BSAS versus EPIC controls gave an adjusted R2 of 0.8683 (Fig. S1).

Bland Altman correlation

A Bland Altman analysis was carried out on the loci investigated by BSAS and compared to data for the same loci produced using the Illumina EPIC array. Figure 1A shows cannabis users (cases) measured by BSAS and the EPIC array on the X axis, while the Y axis represents the log differences between the measurements. The observed differences between loci in cannabis cases (EPIC and BSAS) fall within the lines of agreement. Figure 1B shows the control group differences plotted for the same loci for BSAS and the EPIC array methods. Similar to above, all data points fall within the lower and upper lines of agreement.

Figure 1 Bland Altman of BSAS vs EPIC for cases and controls.

Bland Altman plots showing the log mean differences between DNA methylation as measured by EPIC array vs. the same CpG sites measured using BSAS. (A) Data from cannabis users, gathered using BSAS and the EPIC array (Cases); (B) the control subjects used in BSAS and the EPIC array. Each of the 15 points represents the CpG sites investigated. Dotted lines represent the limits of agreement, red the mean, blue in the 95% confidence intervals.

Mean methylation values for each individual were plotted for each of the 15 loci, and these were then compared between BSAS and EPIC, for cases (Fig. 2) and controls (Fig. 3). Loci displaying a significant shift in average methylation between the methods of detection are indicated with an * when using a Welsh two sample comparison. The following loci were found to display differences between BSAS and EPIC array for cases were: AHRR, cg09078959, cg11293828, CHD7, DP10, EHMT2 and ITPR1, and for controls: AHRR, cg09078959, cg11293828, CHD7, DP10, EHMT2, ITPR1, NIPAL4 and PPM1L.

Figure 2 Individual methylation across the 15 loci for cases.

Average methylation for case individuals only across the 15 loci assessed using EPIC and BSAS. As asterisk (*) represents those loci with significant differences in average methylation between EPIC and BSAS.

Figure 3 Average methylation at the 15 loci for controls.

Average methylation for control individuals across the 15 loci assessed using EPIC and BSAS. As asterisk (*) represents those loci with significant differences in average methylation between EPIC and BSAS.

Assessing amplicon regions

Multiple CpG sites residing within an amplicon can be sequenced using BSAS, providing information about a larger region of interest, rather than just a single CpG site. Fig. S2 displays the multiple CpG sites found across each of the 15 amplicons in this study. A total of 9 of the 15 amplicons contained more than one CpG site. All CpG sites within the amplicons remained non-differentially methylated between cases and controls, except one site in AHRR. The amplicon from SLC17A7 sequenced here contained a total of 15 CpG sites within the 250 base pairs.

Discussion

High throughput array technologies have facilitated the next step in assessing associations between DNA methylation changes in response to a known environmental exposure at a genome-wide level. The EPIC array (as well as the predecessor 27k and 450k arrays) is one such platform that allows for the characterisation of these DNA methylation changes. Through these approaches, various studies have furthered our understanding of how DNA methylation can play a role in response to different environmental exposures.

We selected the orthogonal method BSAS to determine its applicability as a validation, replication and/or expansion tool for EPIC array. BSAS is often used as a standalone method for assessing differential DNA methylation at specific CpG sites, usually because it is more cost-effective than EPIC arrays, and allows analysis of many samples at once, in multiplex. It returns data for all CpGs within a targeted region of interest (∼250 base pairs) with results providing base pair-level specificity (Masser, Stanford & Freeman, 2015). Overall, when considering average methylation between cases and controls as determined via BSAS or EPIC individually, we did not detect significant differences in average methylation for each detection method; the biological results are discussed elsewhere (Osborne et al., 2020), however, it was expected that the smaller sample set used here would not have the statistical power to detect effects found in the larger cohort. The intent of this study was to compare average methylation as determined via BSAS, to that determined by EPIC array. We show here that the estimation of differential DNA methylation observed using BSAS correlated with differential methylation determined via EPIC array. However, although the data correlates between the methods (adjusted R2 cases, 0.8878 and adjusted R2 controls, 0.8683), we urge caution when interpreting this correlation as proof that BSAS will be a suitable independent validation of EPIC array data in every experiment. This is because while the data presented here correlated between BSAS and EPIC array as a whole dataset, some sites showed larger differences between average methylation estimated using BSAS vs. EPIC array. Most notably, where the differential methylation on EPIC array was greater than 5% between cases and controls, BSAS was unable to detect this differential DNA methylation to the same magnitude as EPIC array. Further, a total of 9/15 loci had observed P value significance when carrying out a Welch two sample t-test on control data, and 7/15 on case data, implying there were differences between the methylation values for the methods. For instance, AHRR exhibited a 4% difference in methylation between cases and controls when assessed using BSAS (the highest value detected in using BSAS in this study), compared to 23% using EPIC array. Thus, while a strong correlation between EPIC array data and BSAS data was found across the 15 CpG sites investigated, which itself implies an association between the average methylation at each CpG for the two techniques, each locus must be validated on a case by case basis before being taken forward into high-throughput or large scale screening, to ensure it produces results that are equivalent to EPIC. In addition, further work on CpG sites with higher magnitude changes is needed to determine whether BSAS is limited by the magnitude of differential methylation it is able to detect. However, it is worth noting that most studies of differential methylation report modest (<5%) significant differential methylation observations, suggesting that BSAS may prove useful, given inclusion of rigorous controls of known differential methylation to ensure accuracy of results.

Due to the sequence-based nature of BSAS data (compared to the probe-based nature of EPIC arrays) BSAS, as a standalone method, offers some advantages that are not applicable to EPIC arrays. For instance, BSAS has the potential to determine novel differentially methylated CpGs which may be near (in the same targeted region) but not the initial pre-determined CpG site of interest. This is possible because all CpGs within an, e.g., 500bp region are returned using BSAS data, only one of which may be on an EPIC array. Further, via this targeted sequencing process, BSAS may reveal novel differentially methylated regions (DMRs). DMRs are described as areas which exhibit multiple successive methylated CpG sites which may have biological impact within the genome. For example, they have been implicated in the development and progression of disease (Hotta et al., 2018). Therefore targeting more than a single CpG site may provide further insight into genes and regions of interest. Consequently, while here we have used BSAS technology to replicate/validate differential methylation identified via EPIC array, given that BSAS outputs can correlate with EPIC data, equally, BSAS could be used as a “discovery-based tool”; if significantly differentially methylated CpGs are identified via BSAS, this would serve to justify further investigation using a robust and more expensive high throughout method.

The EPIC array still remains the most reproducible way to measure DNA methylation (Bibikova et al., 2009). This is because the probe-based nature of the method frequently produces comparable results across research groups and arrays. For example, detection of differential methylation using the EPIC array found a difference of 23% in cannabis plus tobacco users, compared to controls, at AHRR (cg05575921, Table 1), a result that is supported by other studies in tobacco smokers using EPIC array (Zeilinger et al., 2013; Guida et al., 2015; Ambatipudi et al., 2016a; Ambatipudi et al., 2016b; Su et al., 2016; Li et al., 2018). AHRR has an important role in controlling a range of different physiological functions; it contributes to regulation of cell growth, regulation of apoptosis and contributes to vascular and immune responses (Trombino et al., 2000; Allan & Sherr, 2005; Lahvis et al., 2005; Marlowe et al., 2008).

BSAS and EPIC array rely upon different chemistries and methods to detect DNA methylation. This may account for the majority of the variation found between the two methods. BSAS relies upon PCR amplification of DNA that is treated with sodium bisulfite. When DNA is treated, unmethylated cytosine residues are converted into uracil via hydrolytic deamination. Amplification of uracil nucleotides during this process are replaced by thymine during replication and the 5-methylcytosines are left unreactive throughout the deamination process and then are amplified as cytosines. It then becomes possible to ‘read’ values of methylation for each cytosine in an amplicon via DNA sequencing (Booth et al., 2013). The ability to treat DNA with sodium bisulfite has led to the expansion of research undertaken within this field (Frommer et al., 1992). However, it is important that we ensure the validity of the results are not limited by the manner in which the data was produced. Ensuring that we limit these discrepancies between technologies will allow for better validation of data. There is potential for errors to occur at this step, because incomplete bisulfite conversion cannot be distinguished from 5-methylcytosine, this can possibly introduce false positive methylation calls at this point (Richards et al., 2018; Krueger et al., 2012). Although both techniques rely upon bisulfite treatment, it is this source of error followed by the PCR amplification that might explain the differences in results we have observed. Refining these sources of error may provide much more comparable results between the two methods.

Conclusions

In conclusion, we chose to validate EPIC array data by using the alternative method, BSAS, to detect differential methylation at CpG sites. While BSAS validated EPIC array data at some loci, and correlated across all loci as a whole, some individual loci did not validate. This implies that each locus must be investigated individually before determining its utility in a large-scale analysis. Further, it is possible that BSAS may be unable to reproduce the magnitude of changes that are shown in the EPIC array system, which may be a consequence of lack of specificity and addition error rate through PCR amplification. It does however, have the ability to assess differentially methylated regions, rather than individual CpG sites. As some regions of the genome are more susceptible to methylation change than others, BSAS could detect swathes of correlated differential methylation at neighbouring CpG sites in certain areas of the genome. From the results shown here, BSAS has the potential to be able to detect methylation marks which maybe hallmarks for disease later on in life. Finally, although BSAS does not generate the same significance level as the EPIC array in some instances, we demonstrate that BSAS can be used as an investigative tool for specific regions of the genome.

Supplemental Information

Supplemental Information 1 CHDS cohort

The Christchurch Health and Developmental Study cohort selected for analysis by BSAS. Cases = cannabis and tobacco users; Controls = never cannabis users.

Click here for additional data file.

Supplemental Information 2 Primer design

Forward and reverse primers used to target validation sites using bisulfite amplicon sequencing CpG sites including an Illumina overhang sequence.

Click here for additional data file.

Supplemental Information 3 Scatter plot with linear regression

Scatter plot with a linear regression of the β values at each locus for BSAS and EPIC array plotted against each other. Colours represent the loci of interest, with the shapes representing the case and controls. There are two regression lines: A represents the correlation between cases with an adjusted R2 = 0.8878 and B represents controls with R2 = 0.8683.

Click here for additional data file.

Supplemental Information 4 All CpG sites assessed using BSAS

Average DNA methylation between cannabis cases compared to controls across all CpGs that were assessed. A differing number of CpG sites are found within each of the 16 gene regions assessed using BSAS. *Cg17470325 wasn’t included in the previous analysis, due to not passing QC using the noob normalisation method for EPIC array analysis. No methylation differences were observed using BSAS for this CpG site either.

Click here for additional data file.

Supplemental Information 5 BSAS raw data

Click here for additional data file.

Additional Information and Declarations

Competing Interests

Author Contributions

Human Ethics

Data Availability

The authors declare there are no competing interests.

Alexandra Noble and Amy J. Osborne conceived and designed the experiments, performed the experiments, analyzed the data, prepared figures and/or tables, authored or reviewed drafts of the paper, and approved the final draft.

J.F. Pearson conceived and designed the experiments, performed the experiments, analyzed the data, authored or reviewed drafts of the paper, and approved the final draft.

Joseph Boden, John Horwood, Neil J. Gemmell and Martin Kennedy conceived and designed the experiments, authored or reviewed drafts of the paper, and approved the final draft.

The following information was supplied relating to ethical approvals (i.e., approving body and any reference numbers):

Collection and analysis of DNA in the Christchurch Health and Development Study was approved by Southern Health and Disability Ethics Committee (CTB/04/11/234/AM10).

The following information was supplied regarding data availability:

R script is available at GitHub:

https://github.com/alexnoble1/Validation-of-EPIC-using-BSAS/blob/master/R%20script.

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
