# Peer review of "A validation of Illumina EPIC array system with bisulfite-based amplicon sequencing"

_PeerJ, doi:10.7717/peerj.10762_

## Round 0.1 · original submission · Major Revisions

The manuscript was examined by two referees with expertise in DNA methylation. Both feel that the subject area of the manuscript is of interest and value to the research community. However, both have adjudged that the current manuscript is unacceptable, with one reviewer suggesting Reject and the other Major Revision decisions. I am going with Major Revision so that additional data presentation and discussion are provided, following which a final decision on publishability can be made. The most significant concern of the reviewers is that only a limited set of data is provided in the manuscript. In the revision, please provide results for full data that was collected. The presentation and discussion of the results should be thorough with all nuances as asked for by the reviewers. I hope you are able to satisfactorily address all of the other comments of the reviewers, including by providing some biological findings in the suggested group comparisons.

Reviewer 1 ·

Basic reporting

The structure of the paper is apporpriate

Experimental design

The authors report a comparison between Illumina EPIC arrays and bisulfite amplicon sequencing. They identified sites associated with smoking using the EPIC array and then attempted to validate 15 of the top sites using bisulfite amplicon sequencing. The results are presented in Figure 1 which shows a scatter plot of the two techniques and Fig 2 which shows a Bland Altman plot.

My primary criticism is that the data as presented in unconvincing. The BSAS approach was carried out on 82 individuals, yet we only see the data for the means of each of the 15 sites in the two groups in Figs 1 and 2. It would be good to show the entire dataset so we can see how reproducible data is across individuals and not just across means.

Fig 2 is problematic The x- axis should be the mean of the measurement of the two approaches but has negative values. Moreover, as the authors state, AHRR has a mean difference of 0.23 in the EPIC array and only 0.04 in BSAS, yet none of the differences in fig 2 are greater than 0.1

Thus it is impossible to determine whether BSAS is actually working or whether the 40 cycles of PCR that are needed to generate sufficient amplicons are significantly biasing the methylation estimates (which could easily happen if the converted and unconverted CpGs have slightly different amplification biases. Moreover, there is no mention of the coverage distribution of each amplicon across individuals. The minimum coverage of 8 seems remarkably low as it limits the resolution of a methylation estimate

Some minor concerns are:

1) In the introduction it is stated that most CpGs are in promoters, which is not correct.

2) When the other methods for targeted DNA methylation profiling are presented, hybridization capture approaches are not mentioned.

3) It is unclear how edge r was used in the analysis, as all we see are figures 1 and 2 that do not involve any estimates of the significance of differential methylation

Validity of the findings

Other bisulfite amplicon approaches are not discussed so it is not clear if there is any novelty

Additional comments

See above

Reviewer 2 ·

Basic reporting

Language, literature references etc. are fine.

Experimental design

The experimental design is ok. However, I miss in depth validation of the BSAS procedure - as mentioned by the authors in line 273 and 274 in the discussion part, they need to investigate, which differences can reliably be detected. that should be done in that manuscript to improve the quality.

Validity of the findings

findings are valid and nice, but I expect more details on the analysis. why did the authors just investigate the correlation between BSAS and the EPIC array for the individual groups and not for the complete data set? why did they not report on results of group comparison etc.
the result section is very weak and should be matter of major revision.

Additional comments

The provided manuscript by Noble et al. aims to investigate an interesting topic, as validation of the EPIC array is of great interest for the community. However the manuscript needs to provide more results tho prove the evidence of the findings. Why did the authors use different sample groups (cannabis/tobacco users vs. non-smokers) and provide just information on the correlation of the individual groups between EPIC and BSAS? they should use this data as a showcas and report also on the biological findings.

Moreover, I guess the authors missed to report the data of all CpGs they have investigated. I am quite sure that they did start with more than 15 CpGs and just picked out the results of the best one.

The manuscript needs more information on the capacity of BSAS. How many samples and CpGs can be investigated e.g. using a Illumina HiSeq or NovaSeq machine?

---

## Round 0.2 · Minor Revisions

Revision 1 of the manuscript was assessed by one of the two reviewers of the original submission. The reviewer has requested changes to two figures. Please address them.

Reviewer 1 ·

Basic reporting

Adequate

Experimental design

Adequate

Validity of the findings

Adequate

Additional comments

I still do not find the figures acceptable.

In figure 1 the x axis is labeled as mean, but presumably represents the log of the mean. Why take the log of the mean in t he first place and if it is deemed necessary then at leas label the figure as log(mean).

In fig 2 it is very difficult to distinguish the results from the epic array and the amplicon data. The different symbols are barely distinguishable. Maybe present them side by instead of showing the on top of each other. It would also be good to show the mean or median of each distribution

---

## Round 0.3 · accepted · Accept

The minor comments raised in the last review have been adequately addressed.